# Responses to Stress: Investigating the Role of Gender, Social Relationships, and Touch Avoidance in Italy

**DOI:** 10.3390/ijerph18020600

**Published:** 2021-01-12

**Authors:** Marcello Passarelli, Laura Casetta, Luca Rizzi, Raffaella Perrella

**Affiliations:** 1Institute of Educational Technology, National Research Council of Italy, 16149 Genova, Italy; 2Associazione Centro di Psicologia e Psicoterapia Funzionale, 35131 Padova, Italy; l.casetta@psicoterapiafunzionale.it (L.C.); l.rizzi@psicoterapiafunzionale.it (L.R.); 3Department of Psychology, University of Campania “Luigi Vanvitelli”, 81100 Caserta, Italy; raffaella.perrella@unicampania.it

**Keywords:** stress response, gender differences, social behavior, attachment, touch avoidance, network analysis

## Abstract

Stress is a physiological response to internal and external events we call “stressors”. Response to the same daily stressors varies across individuals and seems to be higher for women. A possible explanation for this phenomenon is that women perceive sociality, relationships, and intimacy—important sources of both stress and wellbeing—differently from how men experience them. In this study, we investigate how gender, attachment, and touch avoidance predict stress responses on a sample of 335 Italians (216 females; age = 35.82 ± 14.32). Moreover, we analyze the network of relationships between these variables through multiple linear regression and exploratory network analysis techniques. The results recontextualize the role of gender in determining stress responses in terms of (lack of) confidence and touch avoidance toward family members; attitudes toward relationships seem to be the main determinants of stress responses. These results have implications for reducing stress in both clinical settings and at a social level.

## 1. Introduction

Selye was the first to use the term stress to refer to a “response” elicited by a stimulus called a stressor [1,2]. As defined by Everly and Lating, “stress is a physiological response that serves as a mechanism of mediation linking any given stressor to its target-organ effect.” [3,4,5]. 

Stressor events can be psychosocial and biogenic [3]. Psychosocial stressors are either real or imagined environmental events that lay the ground for the elicitation of the stress response. Biogenic stressors, instead, actually “cause” the elicitation of the stress response. Examples of such stimuli include caffeine, amphetamines, and guarana.

Individuals respond differently to the same stressors due to both biological and psychological individual characteristics [6]. In clinical practice, therapists usually intervene in how psychosocial stressors are perceived and managed. 

Analyzing previous studies on influences on stress, it has been found that gender is an important variable in predicting stress responses. Specifically, several studies have highlighted that women, on average, report higher chronic stress and somatic symptoms than men, even when exposed to the same stressors [7,8]. This result has led to diverging interpretations, speculating on the reasons behind this gender difference. One such interpretation of gender differences in stress responses may be found in how males and females behave in social contexts, displaying different behaviors as well as different attitudes regarding social relationships, social touch, and intimacy. 

Since social relationships are a source of both stress and wellbeing, and attachment and intimacy are keystones for mental health for both men and women [9,10], the intention of this study is to provide clinicians with a better understanding of the mechanisms behind the stress-response process so as to improve effective treatments of stress-related disorders and to reduce their cost and length. In particular, in this study, we will investigate the following research questions in the Italian population: 

RQ1—Do gender, attachment dimensions, and attitudes towards touch predict the self-reported intensity of stress responses?

RQ2—How are gender, attachment dimensions, attitudes towards touch, and stress-responses interrelated? 

While RQ1 is a more traditional approach to finding predictors, the exploratory nature of RQ2 has the aim of shedding light on the complex web of relationships between these variables. There is considerable evidence that gender, attachment, attitudes towards touch, and stress responses are interrelated, but the literature is fragmented and studies often consider a single pair of variables. In this study, we will try to gain a bird’s eye view of how individual characteristics interact and influence stress responses, with the aim of guiding the planning of psychological interventions. 

In the following sections, we will discuss the rationale for considering gender, attachment, and attitudes towards touch as important predictors of stress. 

### 1.1. Examining Gender Differences in Social Behaviors from an Evolutionary Perspective 

According to the biosocial model by Wood and Eagly [11], gender differences in social behavior are driven by two factors: physical differences and sociocultural influences. 

Physical differences include women’s ability to bear and nurse children and men’s greater strength, speed, and size, which lead to women lending more parental care than men. Paternity uncertainty and mating opportunity cost hypotheses further explain gender differences in providing paternal care [12]. The former implies that since fathers are never certain that the offspring they are raising are genetically related to them, they perceive higher cost and risk in investing in parental care. The mating opportunity cost hypothesis, additionally, highlights that males that provide extensive parental care incur an opportunity cost since they could invest those resources in mating with a variety of females, thus having more children and spreading their genetic code further [12].

Studies on gender differences in social behavior seem to confirm that men, on average, invest less in their relationships (including nonpaternal relationships). According to an extensive literature review [13], men generally emphasize instrumentality and independence, whereas women value inclusiveness and interdependence. Indeed, males spend more money to elevate their status and favor promotions that benefit their status, suggesting they are more self-oriented as a short-term sexual strategy [12,13]. In contrast, women favor resources that benefit both themselves and others and are more aware of how their actions affect others; they are also more inclined to help others and to use social support to cope with negative emotions [7,14]. These findings suggest that, overall, women are more other-oriented than men [13].

We could, therefore, hypothesize that gender differences in stress originate from what women value most: caring for others. Indeed, minor daily stressors seem to impact women more than men, and the former more often report family- and health-related events that are experienced by other people as being stressful in their environment. On the other hand, men report out-of-family relationships, work, and finances as having a negative impact on their wellbeing [7]. 

### 1.2. How Relationship Bonds Are Created and Maintained: The Role of Touch

Even if social behavior seems to be influenced by gender and driven by different aims, both men and women show the innate need to form and maintain interpersonal relationships [7]. Forming and maintaining social bonds have both survival and reproductive benefits [15]. Humans create bonds with other humans in all cultures; as John Donne [16] stated, “No (person) is an island”. A lack of supportive relationships triggers emotional distress and has harmful effects on the immune system [17,18,19].

Relationships satisfy the “need to belong”, as it has been called by Baumeister and Leary [20]. Not all relationship bonds, however, satisfy this need—only frequent and pleasant interactions with long-term caring and concern do [20]. According to the authors, “many of the emotional problems for which people seek professional help (anxiety, depression, grief, loneliness, relationship problems, and the like) result from people’s failure to meet their belongingness needs”. 

The communication of emotions plays a fundamental role in forming and maintaining long-term relationships, avoiding potentially dangerous individuals, and moving close to those showing positive emotions toward us or need for help [21]. The most important role among emotions to form intimate relationships seem to be played by sympathy and love [22]. Sympathy, sometimes also called compassion [22], is a care-taking emotion that supports other-oriented, altruistic behavior [23]. Love can be intended as referring to romantic love, familial love, and friendship [24]. 

These emotions can be conveyed through several nonverbal channels, including face, body, and touch [25]. Some nonverbal channels seem to be more important than others when communicating specific emotional messages [21]. Touch seems to be especially important for fostering intimacy [26] and exhibiting love and sympathy [21]. Moreover, sympathy seems to be accurately communicated by touch only in dyads involving at least one woman, while anger is accurately communicated by touch only in male-only dyads [26]. Opposite-sex dyads, additionally, seem to more accurately recognize touch-expressed love [25]. 

In addition to gender differences in interpreting touch meaning, we also observe differences in touching behavior: women, compared to men, show more positive attitudes towards and greater willingness to engage in touch with friends and romantic partners [27,28]. However, women are more likely to perceive touch from opposite-sex strangers as unpleasant [27].

In general, the person’s attitude toward touching and being touched has been called touch avoidance [27,28]. Touch avoidance is influenced not only by gender but also by attachment processes during infancy and adulthood [29,30]. Attachment theory provides a useful theoretical framework for understanding the importance of touch in development. Bowlby [31] postulated that human touch facilitates the connection between a child and his caregiver, essential for his wellbeing in the early years and beyond. Other studies have demonstrated that during the first year of life, touch influences the child’s physical and cognitive development [32] as well as his/her health and sense of safeness [33]. Individual differences in adult attachment security have sometimes been conceptualized in terms of categories, such as secure (individuals not anxious about abandonment nor seeking to avoid others); preoccupied (persons anxious over being abandoned but not avoidant in their behavior), dismissing (people with avoidant behavior without anxiety about being deserted), and fearful (individuals both anxious over abandonment and avoiding closeness) [34].

According to attachment theory, early patterns of tactile behavior and the nature of touch between parents and the child can predict the child’s later tendencies to seek or avoid touching people outside the family and attitudes toward touch in adulthood [35]. Indeed, fearful or dismissing individuals avoid and avert touch, unlike secure or preoccupied ones [27,36]. Relationship satisfaction, previous experiences of familial affection, and trust are also positively correlated with mutual touch for romantically involved individuals [37]. 

Summing up, there is ample evidence of gender differences among women and men on attitudes toward touch, the use of touch for conveying emotions, the importance attributed to relationships, and behavior displayed in social life in general. Since women report higher stress than men, in this study, we want to investigate how gender, attachment, and touch avoidance predict stress responses. Moreover, since the literature is fragmented in describing how different variables leading to stress interact among themselves, the second aim of this study is exploring the network of relationships between gender, attachment, touch avoidance, and stress response. 

## 2. Materials and Methods 

### 2.1. Participants and Procedure 

The study was based on a dataset collected for a previous study [38] that involved a total of 335 participants (216 females, 113 males, 6 undisclosed; age = 35.82 ± 14.32, range 16–74) recruited through convenience sampling. We detected no significant gender differences for marital status (Χ^2^(5) = 1.82, *p* = 0.874), education Χ^2^(5) = 8.82, *p* = 0.117), and occupation (Χ^2^(16) = 19.63, *p* = 0.237). Participation was voluntary and anonymous, and participants received no compensation. All tests were completed by pencil-and-paper in a counterbalanced order so as to prevent fatigue and order effects. All missing data (3.5% of responses) were handled through pairwise deletion in all data analyses. This sample size is more than adequate for both multilinear regression (estimated required sample size for 12 predictors, 0.8 power, and moderate effect size (f^2^ = 0.15) is 127) and network analysis since our network is sparse [39]. R codes and data are available on the repository https://github.com/M-Pass/GenderAndStress. 

### 2.2. Measures

#### 2.2.1. Attachment Style Questionnaire 

The attachment style questionnaire (ASQ) [40,41] is a widely used adult attachment measure in both normative and clinical contexts [42]. It comprises 40 items and 5 dimensions. Respondents are asked to indicate their degree of agreement/disagreement with each item on a Likert-type scale ranging from 1 (“strongly disagree”) to 6 (“strongly agree”). The ASQ contained five subscales: (a) Confidence (8 items), (b) Discomfort with Closeness (10 items), (c) Relationships as Secondary (8 items), (d) Need for Approval (7 items), and (e) Preoccupation with Relationships (7 items). Discomfort with Closeness refers to Hazan and Shaver’s [43] avoidant attachment, whereas Relationships as Secondary is consistent with Bartholomew’s [44] concept of dismissing attachment. Need for Approval reflects the respondents’ need for acceptance and confirmation from others, referring to Bartholomew’s [44] fearful and preoccupied attachment. Lastly, Preoccupation with Relationships involves an anxious and dependent approach to relationships, and Confidence (in Self and Others) reflects a secure attachment orientation. The Italian version of the scale has a Cronbach alpha ranging 0.76–0.84 for the five subscales and 10-week retest reliability ranging 0.67–0.78. 

#### 2.2.2. Mesure de Stress Psychologique 

The Mesure de Stress Psychologique (MSP) [45,46] is a 49-item self-report questionnaire that measures responses to stress. Each item has a Likert-type response scale ranging from 1 (“not at all”) to 4 (“to a great extent”). The final score is computed via software, and the higher the score, the higher the level of stress. The items measure different facets of stress responses, asking the respondent to consider thoughts, somatic symptoms, emotions, and behaviors. The Cronbach alpha for the whole scale is 0.94. 

#### 2.2.3. Touch Avoidance Questionnaire 

The touch avoidance questionnaire (TAQ) [27,38] assesses attitudes towards touch in different contexts, such as situations involving romantic partners, family members, friends, professional touch, and touch with complete strangers. The TAQ comprises 37 Likert-type items to which participants are asked to respond on a 5-point scale ranging from 1 (“fully disagree”) to 5 (“fully agree”). The TAQ subscales are Partner (10 items), Same-sex (6 items), Opposite-sex (6 items), Family (6 items), and Stranger (3 items). Higher scores indicate higher touch avoidance (i.e., aversion to touch). The Italian version of the scale has an ordinal alpha ranging 0.59–0.92 for the five subscales and 1-month retest reliability ranging 0.67–0.90. Since TAQ Stranger had an ordinal alpha < 0.70, following a reviewer’s suggestion, it was removed from the analysis. 

### 2.3. Data Analysis Strategy

Our data analysis strategy encompassed both standard multiple linear regression and techniques originally developed in the network analysis framework. This two-fold data analysis approach serves two separate purposes. First, linear regression—a simple, highly constrained model—can help us ascertain which of the variables being considered has the most direct influence on stress responses (as measured by the MSP), investigating RQ1. Second, network analysis can help us explore the complex web of relationships among the variables in a holistic, data-driven way, investigating RQ2. Since we expect several of the measured constructs to be correlated among themselves and to form a complex causal structure, a technique allowing us to observe all relationships at a glance seemed to us the best fit for this highly explorative study.

## 3. Results

### 3.1. Descriptive Statistics and Gender Differences

In Table 1, we report descriptive statistics for the variables considered as well as two-sample t-tests for gender differences. Results show gender differences for viewing relationships as secondary (more common in men) and touch avoidance towards same-sex friends (substantially higher for men). We also observe a gender difference in stress responses bordering significance (t(224,76) = 2.49, *p* = 0.062, Cohen’s d = 0.29). 

### 3.2. Multiple Linear Regression

As the first part of data analysis, we fitted a multiple linear regression model in which stress responses (as measured by the MSP) are predicted by the four touch avoidance factors of the TAQ (after removal of the “Stranger” subscale), the five factors of the ASQ, gender, age, marital status (coded as “being single”), and years of education. The results are summarized in Table 2. Gender has been coded as 0 = female, 1 = male, and adjusted R^2^ for the whole model is 0.36. The only significant predictors of stress responses seem to be two dimensions of attachment: Confidence and Preoccupation with Relationships, with Discomfort with Closeness bordering on significance (*p* = 0.062). All associations are in the expected direction: individuals with less confidence, more discomfort with closeness, and more preoccupation exhibited higher stress responses. Additionally, individuals with fewer years of education reported higher stress responses. Importantly, gender was not a significant predictor in the multiple linear regression model, suggesting that the gender-stress response association is spurious. 

### 3.3. Association Graph

To further explore the relationships among the variables, we computed and plotted a LASSO (least absolute shrinkage and selection operator)-regularized network of partial correlations [39]. Partial correlations are the measure of the residual association between variables X and Y when controlling for all other variables in the network. As such, some variables that appear to be associated when computing 0-order correlations may not be associated when considering partial correlation (as the 0-order correlation is actually spurious). The opposite result is also possible: variables that are associated in the partial correlation network may appear not to be associated when computing 0-order correlations as other variables in the network “mask” their relationship. Partial correlation networks, also called Gaussian graphical models [47] or concentration graphs [48], can be plotted as a weighted network structure in which two variables are connected by an edge if and only if their partial correlation is different from 0 and the edge weight is proportional to the strength of the correlation (ranging from −1 to +1). 

To aid interpretability, such a network can be made sparser using LASSO regularization. Partial correlations are rarely exactly zero due to statistical noise and sampling variability. Therefore, plotting the partial correlation network as-is would result in a graph in which all nodes are connected. A widely employed method to solve this problem and avoid over-interpretation of spurious results is regularizing the partial correlation matrix using the LASSO method [49]. LASSO regularization penalizes the likelihood function used for estimating network edges by limiting the sum of (absolute) partial correlations. This results in the shrinkage of all estimated correlations and, most importantly, the lowest correlations shrink to exactly zero. The extent of regularization is controlled by the tuning parameter λ. A lower λ results in less shrinkage, while a higher λ will set more edges to zero and result in a sparse network. The optimal value for λ can be identified by minimizing the EBIC (extended Bayesian information criterion; [50]), a strategy that tends to select the model that best reproduces the underlying true network structure [51]. Additionally, in the reported graph, we removed all edges with a nonsignificant partial correlation. 

The resulting graph is shown in Figure 1. From the graph, we can see ASQ1, ASQ2, ASQ4, and ASQ5 remain significantly associated with the MSP even after partializing for all other variables (r = −0.19, *p* < 0.001; r = 0.20, *p* = 0.001; r = 0.28, *p* < 0.001; r = 0.14, *p* < 0.001). 

However, the network also provides us a bird’s eye view of all relationships among variables, identifying other important associations. When considering touch avoidance, it is important to note that two factors (touch avoidance towards same-sex friends and opposite-sex friends) are highly related, touch avoidance towards partners is slightly related to touch avoidance towards opposite-sex friends, and touch avoidance towards family members seems to be independent of other measures of touch avoidance. Confidence (ASQ1) is negatively associated to touch avoidance towards family members (r = −0.13, *p* < 0.001) and romantic partners (r = −0.14, *p* < 0.001). Additionally, as could be expected, touch avoidance towards partners is associated with being single (r = 0.14, *p* = 0.001). Notably, while gender is on the periphery of the network, it seems to have a strong relationship with touch avoidance towards same-sex friends (r = 0.053, *p* < 0.001). 

### 3.4. Causal Discovery

One interesting application of network analysis methods is the use of causal discovery algorithms to infer the directionality of associations.

While finding causal relationships from purely observational, cross-sectional data may seem, at first, impossible, clever use of conditional independence can identify, in some cases, the variable that is the cause and the variable that is the effect [52]. This is the basis of the PC-stable algorithm [53]. The algorithm starts by identifying the skeleton of a graph, i.e., the correlation graph. Then, it identifies triplets of variables X, Y, and Z such that (1) Y is associated with X, (2) Y is associated with Z, (3) X and Z are not associated, and (4) X and Z are associated when partializing for Y. For example, in our data, we have such a situation between the variables Gender, TAQ Same Sex, and TAQ Opposite Sex: Gender and TAQ Opposite Sex are both associated with TAQ Same Sex (r = 0.37 and r = 0.61, respectively) but not with each other (r = 0.05). However, if we condition for TAQ Same Sex, the relationship between gender and TAQ Opposite Sex becomes substantial (r = 0.25). In cases such as this, both edges, X—Y and Z—Y, can be oriented towards Y.

This happens because such a triplet of variables has only four possible configurations: X → Y → Z, X ← Y ← Z, X ← Y → Z, and X → Y ← Z. However, only the latter would result in conditional dependence between X and Z when conditioning for Y. By directing those edges, the graph implements new constraints that can be used to further direct other edges in the graph. The algorithm thus proceeds iteratively until it directs all edges for which it is possible to infer directionality.

The resulting graph for our data is reported in Figure 2. Note that some of the weaker relationships (e.g., ASQ3–ASQ4) identified in the partial correlation graph are not included in this graph. This is normal and results from the different estimation process used for determining the graph adjacency matrix. Parameters used for the PC-stable algorithm are α = 0.01, partial correlations as independence tests, majority rule for checking ambiguous edges, and resolution of conflicts via bidirected edges. This particular setting configuration ensures the algorithm is order-invariant.

Before proceeding with the interpretation of the graph, it should be noted that causal discovery algorithms usually have strict assumptions that are likely to be violated in real-world scenarios. One of the most important ones is that no relevant variables and, especially, no common causes should be omitted from the network. Since this assumption is unlikely to be met, results from the PC-stable algorithm should be interpreted very tentatively. However, the directed graph still offers some suggestions and can be used to inform future research studies.

From the graph, we can see that a direction has been identified for most of the relationships. According to this graph, both gender and touch avoidance of opposite-sex friends exert an influence on same-sex touch avoidance. On the other hand, touch avoidance towards family members is associated both to touch avoidance towards same-sex friends and confidence (ASQ1); however, in both cases, the direction of causality is unclear.

The most interesting region of the graph, however, is the long causal chain involving MSP scores. This chain suggests that viewing relationships as secondary increases confidence, which, in turn, decreases stress responses and touch avoidance towards romantic partners. High levels of stress also heighten preoccupation with relationships, which increases the need for approval. MSP scores are also associated with discomfort with closeness (ASQ2), but it is unclear which variable (if any) is the cause and which is the effect.

## 4. Discussion

Our results show that some dimensions of attachment, namely, confidence, discomfort with closeness, and preoccupation with relationships, are closely associated with the self-reported intensity of stress responses. According to the causal discovery algorithm, however, it is likely that the level of confidence causes stress responses and that stress responses cause preoccupation with relationships. Attitudes towards touch are not directly related to stress responses, although we observed a substantial gender difference regarding touch with same-sex friends.

Contrary to what would be expected from previous studies [7,8], we surprisingly found age and gender were not significant predictors of stress responses. Indeed, through the network analysis approach, which we used to explore the interrelation of these variables, gender and age were very much on the periphery in the resulting network of variable associations. Interestingly, from our data, it came to light that the variables more strongly related to stress responses all pertain to attachment style. While attitudes towards touch appear to be indirectly associated with stress responses, it is unclear whether one causes the other. Finally, the resulting model highlights additional relationships between variables, suggesting, for example, that attitudes towards touching friends (of either gender) are strongly associated, while attitudes towards touching romantic partners or family members seem to stem from different psychological processes. The strong association found between gender and attitudes towards touching same-sex friends may be related to cultural norms.

This study has important limitations, and its results should be interpreted with caution.

First, the stress response measure—the MSP—conflates the presence of stressors in one’s life with the reaction towards these stressors. Indeed, the questionnaire asks respondents how often they experience specific unpleasant responses to stress. Reporting high levels of stress can mean that either the respondent lives in an unusually stressful environment or they are unusually sensitive to daily and ordinary stressful events or both. Replicating the results with a more fine-grained measure of stress would aid the interpretation of how dispositional characteristics interact with life events for determining stress responses. Moreover, a laboratory stress procedure would increase the value of the study, overcoming the limits of self-report instruments.

Second, the methods used in this study—regularized partial correlation networks and causal discovery—are highly suited for exploratory studies but can fail to be replicated in subsequent studies. A replication of these findings, perhaps using a more confirmatory analytical approach, would lend more credibility to our findings.

Third, the study has been conducted in Italy. Since attitudes towards relationships and, especially, social touch differ between cultural contexts [54], future lines of research should focus on testing whether the same causal structure between variables can be observed among the population of other countries.

The causal discovery approach allowed us to find a model that can be used to inform psychological interventions for reducing stress responses, although it bears restating that these results may fail to be replicated and should be interpreted very tentatively. Specifically, the main suggestion emerging from our data is that interventions in clinical settings should focus on improving confidence. One possible way to do so is helping individuals deprived of caring touch during childhood to experience caring and affectionate touch, teaching them to communicate and recognize love and sympathy through touch. Possible approaches for teaching caring touch include compassion-focused therapy [55], in which caring touch is visualized through guided imagery, or functional psychotherapy, in which caring touch is actually employed as a relational tool by a therapist [54]. A second way to improve confidence is helping both men and women strike a balance between their own needs and the needs of their significant others. Mindful self-compassion training, which focuses on the recognition and validation of one’s own needs, offers useful tools in this regard [56]. Additionally, our model suggests some important benefits of reducing stress responses, such as improving comfort with closeness and reducing preoccupation with relationships. As a consequence of this result, suggesting client homework such as being massaged by their partner, which could improve their comfort with closeness, could be ineffective if we do not reduce the stress response first. Moreover, intervening on reducing preoccupation with relationships might follow an intervention based on building a sense of confidence and safeness. Gender, instead, appeared to be a mostly inconsequential variable in the network, apart from its strong relationship with same-sex touch avoidance (likely due to social norms). Therefore, gender norms and identity should not be the focus in stress reduction interventions as they are unlikely to play a substantial role.

Lastly, this study also confirms the importance of interventions focused on prevention at a social level to facilitate a secure attachment between parents and children. Since children learn primarily by imitating adults, it would be useful to offer parental training on emotional intelligence and psychological flexibility, with the aim of providing tools for recognizing, validating, and communicating emotions efficiently so as to build pleasant bonds with children in a caring environment [57]. Another important intervention in prevention should consider how to reduce the incidence of postpartum depression. It is well known that depressed mothers report bonding difficulties with their children [58]. Social support and stress interventions during pregnancy have been validated in preventing postpartum depression [59], and efficient interventions have been described to reduce postpartum depression and improve mother–children interactions [60]. However, Italy still lacks a standardized procedure to care for and assess maternal psychological health despite its centrality to parents’ and children’s wellbeing in the short and long term. In conclusion, our analysis recontextualizes the role of gender in determining responses to stress, suggesting that when we also consider attachment styles and attitudes towards closeness and intimacy, gender appears not to be as central as it would appear at first glance. Interventions in a clinical setting for reducing stress might first consider the client’s confidence instead of gender. Moreover, we believe that early prevention could reduce the incidence of stress-related diseases throughout the life-cycle. It should be noted, however, that this study considered an Italian sample and that the relationship between gender, attachment dimensions, attitudes towards touch, and stress responses may differ in other cultural contexts. This is especially true since different genders face different challenges according to societal norms [61], and attitudes towards touch have great variation between cultures [62,63].

## 5. Limits

The study has some limitations. First of all, the sample is unbalanced (34% males). While the methods used are robust to unbalanced samples, additional data on men could shed more light on the relationship between attachment, stress responses, and touch avoidance.

More importantly, the questionnaire used to assess stress level is ambiguous in what it measures (stressful contexts, stress response attitudes, or both). A more in-depth exploration of stress responses and stressors, based on a sound theoretical framework, could improve our understanding of how gender relates to stress.

Lastly, a more general word of caution is that the methods used (regularized partial correlation networks and causal discovery) to analyze the data are highly suited for exploratory studies but can fail to be replicated in subsequent studies. Additionally, the study was conducted in Italy, so the results cannot be generalized to other countries.

## 6. Conclusions

This study started from the consideration that women report higher stress than men, even when facing the same stressful events [7,8]. Furthermore, women experience relationships—a major source of both stress and wellbeing—differently from how men experience them, a claim backed by evolutionary psychology [12,13]. Using a highly exploratory design, we investigated in an Italian sample to see if stress responses were predicted by gender, attitudes towards intimacy and closeness (as measured by the attachment Style questionnaire), and attitudes towards interpersonal touch (as measured by the touch avoidance questionnaire). Moreover, we investigated the network of relationships between these variables with the intention of integrating research results that were previously fragmented.

Our results surprisingly recontextualize the role of gender in determining stress responses, as (lack of) confidence, touch avoidance toward family members, and attitudes toward relationships seem to be the main determinants of stress responses.

Overall, our study has important implications in both clinical settings and at a social level. First, clinicians might use these findings and intervene on touch attitudes and lack of confidence to reduce the stress response at its roots. Secondly, our study underlines the importance of prevention at a social level, in particular, psychologically supporting parents from pregnancy throughout the first years of their children’s lives in order to build more secure and strong bonds.

Future research should investigate the same topic with different instruments to measure stress response, in particular, employing physiological variables such as heart rate variability or salivary cortisol. Moreover, it would be interesting to evaluate the effectiveness of specific psychological approaches in intervening in the social and relational sphere, comparing their efficacy with those intervening in other spheres such as thoughts, beliefs, and symptoms. These studies could help practitioners improve the efficacy of psychological interventions and reduce their costs and length.

## Figures and Tables

**Figure 1 ijerph-18-00600-f001:**
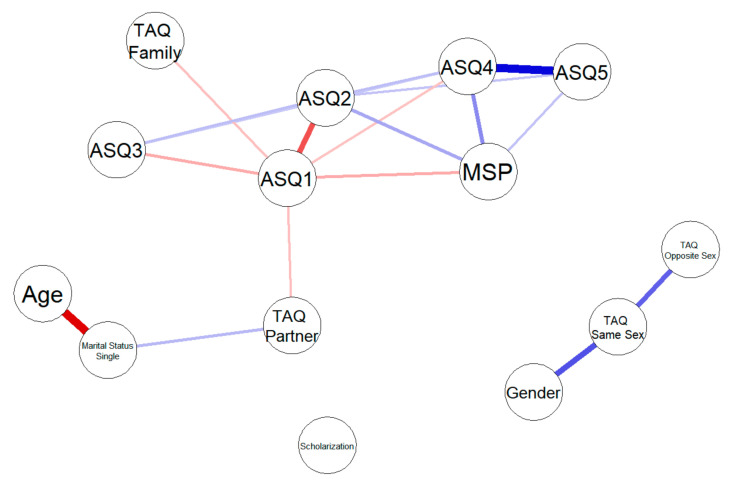
LASSO (least absolute shrinkage and selection operator)-regularized network of partial correlations. Blue lines are positive correlations; red lines are negative correlations. Line width is proportional to partial correlation magnitude.

**Figure 2 ijerph-18-00600-f002:**
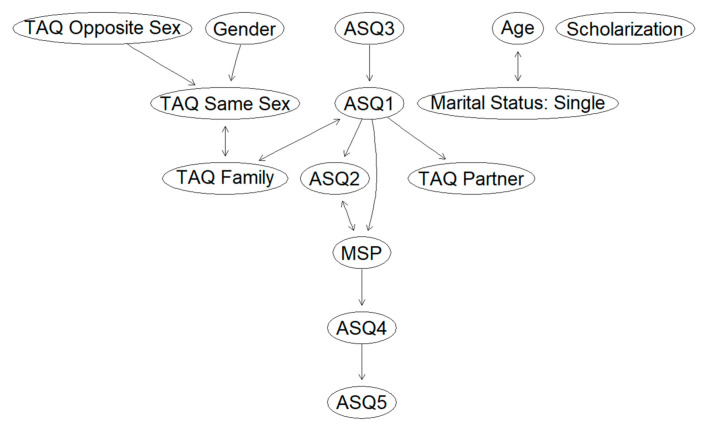
Directed network of the variables’ relationships.

**Table 1 ijerph-18-00600-t001:** Descriptive statistics for measured variables and *t*-tests for gender differences. ** = significant for *p* < 0.01; *** = significant for *p* < 0.001. All p-values have been adjusted using Benjamini–Hochberg’s correction for multiple comparisons.

Variable	Mean and Standard Deviation	Mean and Standard Deviation (Men)	Mean and Standard Deviation (Women)	*t*-Test for Gender Differences
MSP (Stress responses)	88.86 ± 23.12	84.43 ± 20.39	91.21 ± 24.17	t(224.76) = 2.49, *p* = 0.062, Cohen’s d = 0.29
ASQ1 (Confidence)	32.41 ± 5.15	32.48 ± 4.70	32.39 ± 5.38	t(250) = −0.16, *p* = 0.939, Cohen’s d = 0.02
ASQ2 (Discomfort with Closeness)	35.13 ± 7.94	34.29 ± 6.70	35.54 ± 8.51	t(260.64) = 1.41, *p* = 0.375, Cohen’s d = 0.16
ASQ3 (Relationships as Secondary)	12.89 ± 4.33	14.01 ± 4.66	12.86 ± 4.01	t(186.15) = −3.24, *p* = 0.010 **, Cohen’s d = 0.41
ASQ4 (Preoccupation with Relationships)	20.42 ± 5.81	20.06 ± 5.37	20.59 ± 6.04	t(243.08) = 0.81, *p* = 0.587, Cohen’s d = 0.09
ASQ5 (Need for Approval)	23.19 ± 5.19	22.73 ± 5.03	23.43 ± 5.28	t(226.41) = 1.16, *p* = 0.436, Cohen’s d = 0.13
TAQ—Same Sex	2.07 ± 0.74	2.47 ± 0.81	1.86 ± 0.60	t(117.8) = −7.16, *p* < 0.001 ***, Cohen’s d = 0.91
TAQ—Opposite Sex	2.16 ± 0.74	2.10 ± 0.64	2.18 ± 0.79	t(269.95) = 1.04, *p* = 0.463, Cohen’s d = 0.11
TAQ—Family	2.65 ± 0.96	2.75 ± 0.89	2.60 ± 0.99	t(249.37) = −1.41, *p* = 0.375, Cohen’s d = 0.16
TAQ—Partner	1.94 ± 0.60	1.89 ± 0.53	1.97 ± 0.63	t(257.03) = 1.23, *p* = 0.436, Cohen’s d = 0.14
Age	35.82 ± 14.32	36.40 ± 14.57	35.56 ± 14.23	t(222.81) = −0.50, *p* = 0.746, Cohen’s d = 0.06
Years of scholarization	14.95 ± 3.35	14.59 ± 3.05	15.13 ± 3.50	t(255.75) = 1.44, *p* = 0.375, Cohen’s d = 0.16

MSP = Mesure de Stress Psychologique; ASQ = Attachment Style Questionnaire; TAQ = Touch Avoidance Questionnaire.

**Table 2 ijerph-18-00600-t002:** Results of the multiple linear regression model. Df = 226. ** = significant for *p* < 0.01. All *p*-values have been adjusted using Benjamini–Hochberg’s correction for multiple comparisons.

Predictor	Std. β	t	*p*-Value
(intercept)	0.86	3.26	0.006 **
ASQ1 (Confidence)	−0.24	−3.59	0.006 **
ASQ2 (Discomfort with closeness)	0.16	2.34	0.057
ASQ3 (Relationships as Secondary)	−0.09	−1.56	0.187
ASQ4 (Preoccupation with Relationships)	0.24	3.38	0.006 **
ASQ5 (Need for Approval)	0.10	1.43	0.196
TAQ—Same Sex	−0.10	−1.48	0.196
TAQ—Opposite Sex	0.00	−0.03	0.976
TAQ—Family	0.06	1.10	0.318
TAQ—Partner	0.05	0.93	0.383
Gender	−0.11	−1.76	0.140
Age	−0.12	−1.91	0.121
Marital status (single)	−0.25	−1.89	0.121
Years of education	−0.05	−3.18	0.006 **

MSP = Mesure de Stress Psychologique; ASQ = Attachment Style Questionnaire; TAQ = Touch Avoidance Questionnaire.

## Data Availability

Publicly available datasets were analyzed in this study. This data can be found here: https://github.com/M-Pass/GenderAndStress.

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
