# Peer review of "Responses to Stress: Investigating the Role of Gender, Social Relationships, and Touch Avoidance in Italy"

_ijerph, 2021, doi:10.3390/ijerph18020600_

Round 1

Reviewer 1 Report

As I wrote in my previous review, the problem of individual response to stress is of particular relevance and importance in the present unstable time. And now I can say that over the past few months the urgency of the problem under study has not diminished, but increased. Also, I must emphasize that the authors have done a lot of work to improve the article. They took into account and corrected all the comments I made in the first two reviews.

This version of the article has a title that corresponds to the research questions, to the structure of the article, and to content of the sections and subsections. I really like the revised content of the Discussion and Conclusions sections, I believe that these sections now fully comply with the rules for scientific articles.

Maybe it needs a little editing, including English, but this is up to the Editor’s decision.

Author Response

Thank you (again!) for your positive review. 

Reviewer 2 Report

The authors have made a few small changes and have resubmitted the article. However, it still has the same problems as in the previous version. That is:

Regarding the sample, it would be necessary to include additional characteristics to control their potential confounding effect in the results. That is, to include demographics as covariates.

There is a considerable imbalance in gender distribution of the sample. As ‘gender’ is an important factor for this study, it would be necessary to include a more balanced sample

I still find it suspicious that three questionnaires need to be counterbalanced to avoid fatigue. This makes me think that the authors have extracted these data from a larger study and have taken advantage of a small one. This is a common practice known as "slicing salami."

Regarding psychometric properties of questionnaires, it is important to specify the Cronbach alpha for each subscale. In fact, those subscales with values lower than .70 are a potential limitation and it would be important to remove from analysis. In fact, they have omitted in this version that one of the TAQ subscales presented a Cronbach alpha with a .64 value. Therefore, it would be necessary to include information of each subscale and to remove those subscales with lower values than .70.

After Bonferroni correction a p value of .013 is not significant. Hence, this would affect the interpretation of your results.

Finally, there is a considerable number of published papers assessing the association of these variables. Hence, it would be interesting to include biological markers or new instruments to assess this field of research. This is why it would be necessary a new point of view of crossing self-reports with laboratory assessments.   

Author Response

Here are our point-by-point responses to the issues raised by the reviewer:

Regarding the sample, it would be necessary to include additional characteristics to control their potential confounding effect in the results. That is, to include demographics as covariates.

Gender and age have already been included as covariates in the multiple regression, and have already been included in the network. Occupation and highest school level completed cannot be included in the network, as they are categorial variables. While we could dummy-code them, we believe the result would be confusing, and it's unclear to us - on a theretical standpoint - why these variables would be relevant for investigating attitudes towars touch or stress responses. 

We do agree that marital status could play a significant role, and we included it in the revised version of the manuscript. 

There is a considerable imbalance in gender distribution of the sample. As ‘gender’ is an important factor for this study, it would be necessary to include a more balanced sample

While having more data on men would be valuable, the methods we employed are robust to unbalanced samples (and 34% is not that unbalanced to begin with). 

I still find it suspicious that three questionnaires need to be counterbalanced to avoid fatigue. This makes me think that the authors have extracted these data from a larger study and have taken advantage of a small one. This is a common practice known as "slicing salami."

There is no salami slicing going on here -- and, frankly, your insistence on this point makes us wonder if the review is subject to undue prejudice. Please, refrain from casting unfounded accusations. Counterbalancing is standard practice. 

Regarding psychometric properties of questionnaires, it is important to specify the Cronbach alpha for each subscale. In fact, those subscales with values lower than .70 are a potential limitation and it would be important to remove from analysis. In fact, they have omitted in this version that one of the TAQ subscales presented a Cronbach alpha with a .64 value. Therefore, it would be necessary to include information of each subscale and to remove those subscales with lower values than .70.

We already included the alpha for each scale used. We argued in response to your previous review for our decision to keep the single subscale with alpha < .70. However, following your suggestion, we removed this subscale from the analysis and ran the analyses without it. This changed results somewhat (mostly in the direction of making touch avoidance less central to the network), which in turn changed our conclusions. 

After Bonferroni correction a p value of .013 is not significant. Hence, this would affect the interpretation of your results.

As we explained is our previous response to reviewers, the p-value of .013 has already been corrected for multiple comparisons. That is, the uncorrected p-value was higher (.0018, to be precise), and, after applying correction for multiple comparisons, the p-value was .013. 

Do note that we applied Benjamini-Hochberg's correction, not Bonferroni's. Bonferroni correction is overly conservative and considered deprecated (despite still being taught in undergraduate courses). 

However, in the revised version of the manuscript, you won't see the value .013, since the addition of marital status and the removal of TAQ Stranger changed the estimates of the linear regression model. 

Finally, there is a considerable number of published papers assessing the association of these variables. Hence, it would be interesting to include biological markers or new instruments to assess this field of research. This is why it would be necessary a new point of view of crossing self-reports with laboratory assessments.   

As argued in the conclusions and in our previous response, we agree that this would be an interesting direction for future studies.

Reviewer 3 Report

I greatly appreciate the opportunity to review research on gender issues from a psychological and social perspective. The work presented, however, does not appear to be a manuscript to be evaluated, but a draft; a revised, modified, but not prepared document for a reviewer to do their job of evaluating. Authors are suggested to be more careful with questions of form like this. Once these questions are reviewed, it would be possible to do the review work.

Author Response

Unfortunately, as this is a re-submission after extensive revisions, we used the "track changes" functionality of Word. When uploading the file as PDF, the revisions look like comments, which may give the misleading impression that this is a draft. It is not a draft. 

Round 2

Reviewer 2 Report

The authors have tried to adapt the commentaries that we made in previous reviews. However, the main problems this manuscript remains as in previous ones. That is, their main objective is not new as there are a considerable number of studies assessing that.

It would be nice to incorporated new instruments because three questionnaires do not seem enough to me to analyse stress perception.

Due to the characteristics of the study, I think it would be good if they expanded the sample and employed men.

I think they will incorporate more demographic variables (e.g., educational level, working status, number of children…) and specific variables regarding mental and physical health. In fact, I consider that it would be important to consider these variables as potential moderating and/or mediating variables.

What are the clinical applications of this study?

Moreover, the authors have pointed out the limitations of the instruments used in the study. Therefore, this reduces the interest and importance of the study.

Author Response

We added years of scholarization into the regression and the network, so as to provide more information regarding how demographic characteristics interact with the variables examined. Note that this changes some p-values around, due to correction for multiple comparisons. Some results have therefore changed slightly, although most of them remained the same (most importantly: the causal network is the same). We also expanded the considerations on clinical applications of the study. 

The other requests are non-fixable without running a new study, which at the moment would be unfeasible. This is a very peculiar time for measuring attitudes towards touch, given the exended social distancing. Any result we would find now on the relationships between touch, attachment and gender would likely be polluted by the contingent, exceptional situation. 

Apart from this -- we understand that the reviewer doesn't find the study interesting, as they expressed this sentiment time and time again during the reviewing process. However, this is a matter of editorial policy, and doesn't pertain to the study quality. 

This manuscript is a resubmission of an earlier submission. The following is a list of the peer review reports and author responses from that submission.

Round 1

Reviewer 1 Report

The problem of individual response to stress is of particular relevance and importance in the present unstable time. The authors of this article examine gender differences in stress response depending on attachment style and attitudes towards touch. In my opinion, such a consideration of the stress response problem has novelty, originality and practical significance.

Unfortunately, despite the good theoretical substantiation and the use of modern statistical methods in the research, its presentation in the article has a number of significant drawbacks.

  1. There aren’t a clear aim and hypotheses in the article. For example, the authors report on "gender differences" in the title, on "the association between stress responses, gender, attachment, and attitudes towards touch" in the annotation, and on “explaining gender differences in stress response by the different attitudes exhibited by men and women towards social life, relationships, and touch” in the Introduction. These formulations are similar, but different aspects of the problem under study. Therefore, the article needs to clearly identify the aim, research questions and hypotheses.
  2. I believe that the description of the data analysis strategy and statistical methods should be moved to section 2, subsection 2.3.
  3. In the "Results" section, the authors provide only data of standard multiple linear regression and network analysis. However, they do not provide descriptive statistics for the studied variables and their differences between men and women, which is necessary for an article on gender differences. Also, I think it would be helpful to fit the regression models separately for men and women and then compare them.
  4. After the corrections described above, the authors will need to clarify the content of the "Discussion" and "Conclusions" sections. In the presented version of the article, "Discussion" is more like conclusions, and "Conclusions" contain only research limitations.
  5. The authors are aware that a major limitation of the study is that it was conducted in Italy (lines 325 – 328), but I believe that this important fact should be reflected in the title of the article, as well as in the discussion of the study results.

Thus, the authors discuss an important and relevant problem, but the article needs significant revision in order to more fully and correctly presents the results of the study to the readers.

Reviewer 2 Report

Although this is an interesting article, I consider that there are a considerable number of methodological limitations that the authors must resolve.

Firstly, the authors claim that the article present a "more than adequate" sample size, but I would like to know the exact recommended sample size based on the proposed characteristics. In this sense, due to the characteristics of the study (e.g. cross-sectional nature, absence of biological samples as correlates of stress, absence of a standardized laboratory procedure, etc.). Therefore, it would be necessary to consider a higher power, which increase the need of a higher sample size close to 400 participants. Also, what do the authors consider as moderate effect size? It would be necessary to the exact value. Additionally, it would be necessary to error that authors consider appropriate to run their study.

Regarding the sample, it would be necessary to include additional characteristics to control their potential confounding effect in the results.

There is a considerable imbalance in gender distribution of the sample. As ‘gender’ is an important factor for this study, it would be necessary to include a more balanced sample. Furthermore, authors have included participants from 16 to 74. In this sense, there are several variables to might interfere with the ability to cope with stress of each participants. For example, I think that the worries of an individual with 20 years old is completely different of an individual with 65 years old.

It would be interesting to include the main reason for counterbalancing three self-reports administration.

As it is important for this study to assess gender differences to cope with acute stress, it would be necessary to include a laboratory stress procedure to complement self-reports results. This would increase the value of the manuscript.

There is an absence of information in the manuscript. For example, it would be necessary to include reliability and validity of self-reports.

It would be necessary to include the ‘data analysis’ in the ‘methods’ section.

It is an important statistical limitation, which is the absence of multiple comparisons control (e.g. Bonferroni correction).

Finally, based on the methodological weakness of the study, authors have assumptions and present results as ‘casual’, but it would be important to temperate their sentences, specially in the ‘abstract’. For example, ‘Response to daily stressors varies across individuals and seems to be higher for women than for men.’ This statement is very broad. In fact, more concrete affirmations and supported by concrete evidence would be necessary. Furthermore, it would be recommendable to provide more information about what you consider to be "stress response", does it refer to the ability to cope with stress?

Round 2

Reviewer 1 Report

Dear Authors,

You have done a good job of revising the article. I appreciate that you took into account my recommendations.

Due to the fact that you changed the article title (removed “gender differences”) and clearly formulated the research questions, I withdraw my comment about regression models for men and women separately.

I have only few small recommendations:

  • I recommend to include a brief description of the participants in the Abstract;
  • I propose to include a few sentences after Table 1 describing its results. For example, the sentence from the next subsection 3.2 (“It should be noted that we do observe a gender 216 difference in stress responses bordering significance (t(224,76) = 2.49, p = .053, Cohen’s d = .29”), in my opinion, should be here. By the way, the fact that there aren’t significant gender differences between the studied variables in your sample, which contradicts the data of many previous studies, in our opinion, requires additional discussion;
  • I still believe that "Discussion" section is more like conclusions, and "Conclusions" contain the research limitations and prospects, but no conclusions. However, I do not insist and I trust the decision to the Editor.

Author Response

Thank you for the positive review. Here are point-by-point answers to the remaining issues you pointed out:

  • I recommend to include a brief description of the participants in the Abstract;

We added a brief description of the sample to the abstract.

  • I propose to include a few sentences after Table 1 describing its results. For example, the sentence from the next subsection 3.2 (“It should be noted that we do observe a gender 216 difference in stress responses bordering significance (t(224,76) = 2.49, p = .053, Cohen’s d = .29”), in my opinion, should be here. By the way, the fact that there aren’t significant gender differences between the studied variables in your sample, which contradicts the data of many previous studies, in our opinion, requires additional discussion;

We added the requested information to the manuscript, as well as further highlighting observed gender differences.

  • I still believe that "Discussion" section is more like conclusions, and "Conclusions" contain the research limitations and prospects, but no conclusions. However, I do not insist and I trust the decision to the Editor.

We appreciate your concern, and we revised the Discussions and Conclusions accordingly.

Reviewer 2 Report

The authors have reinforced their article by including new information or clarifying previously included information. However, I’m still concerned due to the fact that revision have revealed serious issues as well as new data analyses have affected the findings and interpretation. Moreover, this manuscript is to

Firstly, it would be important to consider larger sample size based on the characteristics of the study (three questionnaires). In fact, the authors need larger statistical power instead of maintained values of previously version.

Secondly, the importance of consider demographic variables that in the scientific are important for the ability to cope with stress. In fact, there are important these variables for their relationship with stress not with gender.

Moreover, I’m still concerned about the need to counterbalance three questionnaires to prevent fatigue. Counterbalancing would be important to a larger set of questionnaires, but it is weird to counterbalancing three questionnaires.

Regarding psychometric properties of questionnaires, it is important to specify the Cronbach alpha for each subscale. In fact, those subscales with values lower than .70 are a potential limitation and it would be important to remove from analysis.

With regards to analysis, a higher p value than .50 is not a significant result. Therefore, there was not significant gender differences in stress responses (p. 6).

After Bonferroni correction a p value of .013 is not significant. Hence, this would affect the interpretation of your results.

Finally, the absence of biological markers of other assessment of copying with stress diminish my interest for this article. In fact, there is a larger scientific literature assessing gender differences to cope with stress. This is why it would be necessary a new point of view of crossing self-reports with laboratory assessments.    

Author Response

  • Firstly, it would be important to consider larger sample size based on the characteristics of the study (three questionnaires). In fact, the authors need larger statistical power instead of maintained values of previously version.

Our first revision of the manuscript contains a detailed power analysis, as well as considerations regarding sample sizes for network analysis (see 10.3758/s13428-017-0862-1). Power analysis does require some assumptions regarding parameters, and we understand that there is no simple answer to the question “which values should we choose for power analysis parameters?”.

However, while we are willing to revise our power analysis, we would like to keep the discussion grounded on parameters choice. A statement such as “a study with three questionnaires requires a larger sample size” is excessively vague and based on thumb rules and approximation. Power analysis, on the other hand, while debatable in its application, gives at least a concrete frame of reference.

  • Secondly, the importance of consider demographic variables that in the scientific are important for the ability to cope with stress. In fact, there are important these variables for their relationship with stress not with gender.

While we appreciate that there are several variables that are associated with stress (most notably, socioeconomical status), they are not the focus of this study and were not collected in our sample. Regardless, we believe the study to be of interest for IJEPRH and, specifically, for the special issue on women’s health.

  • Moreover, I’m still concerned about the need to counterbalance three questionnaires to prevent fatigue. Counterbalancing would be important to a larger set of questionnaires, but it is weird to counterbalancing three questionnaires.

As argued in the previous revision, counterbalancing is standard practice when collecting data via questionnaires. We agree that it’s probably unnecessary when dealing with only three questionnaires, but it’s not actively harmful.

  • Regarding psychometric properties of questionnaires, it is important to specify the Cronbach alpha for each subscale. In fact, those subscales with values lower than .70 are a potential limitation and it would be important to remove from analysis.

Our dataset does measure a variable with (ordinal) alpha < .70; namely, the TAQ Stranger subscale. This subscale has only three items, which may be the reason alpha is this low. We voiced your same concern when we validated the TAQ in Italian (10.3389/fpsyg.2020.01673). Quoting from that article:

For the TAQ, we observe an ordinal α = 0.84 (0.81, 0.87) for the Partner subscale, […] and ordinal α = 0.59 (0.54, 0.70) for the Stranger subscale. The low value of α for the latter subscale may be because the subscale consists of only three items and α is sensitive to scale length. However, Spearman–Brown “prophecy” formula (Brown, 1910; Spearman, 1910) would predict an α as low as 0.74 were to subscale to comprise six items, suggesting that the Stranger subscale does have relatively low internal consistency. Ordinal α for the whole TAQ is 0.89 (0.85, 0.91). For comparison, in the original validation paper, αs were 0.86 (Partner), 0.89 (Same Sex), 0.85 (Opposite Sex), 0.85 (Family), and 0.64 (Stranger)

However, reliability estimation via test-retest yielded excellent results (r = .89 after one week, r = .82 after one month), which was one of the reasons we kept the subscale despite the suboptimal ordinal alpha.

If the editor believes this subscale should be removed from the paper, we are willing to re-run the analysis without the Stranger TAQ subscale.

  • With regards to analysis, a higher p value than .50 is not a significant result. Therefore, there was not significant gender differences in stress responses (p. 6).

The p-value is .053. It’s not significant, and we do not claim that it is; however, it’s close enough to .05 that we believe we should mention that it “borders significance” (lines 205-206 of the second revision)

  • After Bonferroni correction a p value of .013 is not significant. Hence, this would affect the interpretation of your results.

The value of .013 has already been corrected for multiple comparisons using Benjamini-Hochberg correction. Scientific consensus is that Bonferroni correction is excessively conservative, and Benjamini-Hochberg correction leads to less biased adjustments (10.1016/j.jclinepi.2014.03.012, 10.1111/j.2517-6161.1995.tb02031.x).

  • Finally, the absence of biological markers of other assessment of copying with stress diminish my interest for this article. In fact, there is a larger scientific literature assessing gender differences to cope with stress. This is why it would be necessary a new point of view of crossing self-reports with laboratory assessments.    

As noted in the previous revision – we agree that physiological measures would improve the study, and we noted that in the future directions. However, we believe that self-report-based research still has a role to play and a strong tradition. We do not believe the use of self-reports only to be a critical flaw of our study.